# The Influence of Magnetic Field and Nanoparticle Concentration on the Thin Film Colloidal Deposition Process of Magnetic Nanoparticles: The Search for High-Efficiency Hematite Photoanodes

**DOI:** 10.3390/nano12101636

**Published:** 2022-05-11

**Authors:** Murillo Henrique de Matos Rodrigues, Joao Batista Souza Junior, Edson R. Leite

**Affiliations:** 1Department of Chemistry, Federal University of São Carlos, Via Washington Luiz, km 235, São Carlos 13565-905, SP, Brazil; murillo.matos@lnnano.cnpem.br; 2Brazilian Nanotechnology National Laboratory (LNNano), Brazilian Center for Research in Energy and Materials (CNPEM), Campinas 13083-970, SP, Brazil; joao.junior@lnnano.cnpem.br

**Keywords:** magnetite, dip coating, hematite photoanodes, photoelectrochemical measurements

## Abstract

Hematite is considered a promising photoanode material for photoelectrochemical water splitting, and the literature has shown that the photoanode production process has an impact on the final efficiency of hydrogen generation. Among the methods used to process hematite photoanode, we can highlight the thin films from the colloidal deposition process of magnetic nanoparticles. This technique leads to the production of high-performance hematite photoanode. However, little is known about the influence of the magnetic field and heat treatment parameters on the final properties of hematite photoanodes. Here, we will evaluate those processing parameters in the morphology and photoelectrochemical properties of nanostructured hematite anodes. The analysis of thickness demonstrated a relationship between the magnetic field and nanoparticles concentration utilized to prepare the thin films, showing that the higher magnetic fields decrease the thickness. The *J_abs_* results corroborate to influence the magnetic field since the use of a higher magnetic field decreases the deposited material amount, consequently decreasing the absorption of the thin films. The PEC measurements showed that at higher concentrations, the use of higher magnetic fields increases the *J_PH_* values, and lower magnetic fields cause a decrease in *J_PH_* when using the higher nanoparticle concentrations.

## 1. Introduction

With the development of hematite (α-Fe_2_O_3_) photoanodes for photoelectrochemical cell (PEC) devices, the use of inexpensive, naturally abundant, and electrochemically stable materials can become a reality [1,2]. Recently, considerable progress was made in the improvement of hematite photoanodes for promoting water photoelectrolysis via sunlight [3]. For instance, for a columnar hematite photoanode, a photocurrent as high as 6 mA cm^−2^ was reported [4]. However, despite recent advancements, further research is essential to extract the full potential of hematite photoanodes and establish this material as a viable alternative for producing hydrogen (H_2_) by applying PEC water splitting.

The technique of tuning the nanostructure of hematite thin films to enhance PEC performance is documented in the literature [5]. Although several methods for producing hematite thin films are reported, the colloidal nanoparticle deposition (CND) technique has demonstrated the most promising results [6,7]. In contrast to methods that require an iron precursor or compound (molecular or salt compounds) for producing the hematite phase above a transparent conducting oxide (TCO) substrate, such as atmospheric pressure chemical vapor deposition (APCVD) [8], atomic layer chemical vapor deposition [4], the Pechini method [9], electrochemical deposition [10], and hydrothermal synthesis, the CND method utilizes iron oxide nanoparticles in the preparation of PEC. This process generally involves the thermal treatment of nanoparticles to generate a hematite structure, followed by adhesion to the TCO substrate. The overall process involves three steps: the synthesis of the iron oxide colloidal solution, the deposition of nanoparticles onto the TCO substrate, and high-temperature sintering. Iron-oxide nanoparticles of different sizes, shapes, size distributions, and surface chemical compositions can be synthesized using various techniques. In addition, the crystallographic structure of iron oxide nanoparticles can undergo a phase transformation from Fe_3_O_4_ (${\mathrm{Fe}}^{+2}O,{\mathrm{Fe}}_{2}^{+3}O_{3}$, magnetite) to γ-Fe_2_O_3_ (maghemite). This adaptability in adjusting the iron oxide precursor for the production of hematite PEC makes the CND method a candidate for achieving the desired maximum photocurrent extraction for water splitting. The obtained colloidal solution can be deposited using various techniques, typically spin or dip coating, and the final PEC is obtained by thermally treating the nanoparticles as deposited.

Sivula et al. [7] were the first to prepare hematite photoanode thin films using the CND method and reported a strong dependence of the photocurrent on the sintering temperature. Gonçalves et al. [6] developed an alternative method for preparing hematite photoanodes using the CND process. Magnetite nanocrystals were used as a precursor to producing a hematite photoanode by oxidizing the magnetite structure to hematite at high temperatures. After sintering was performed at 820 °C, the hematite photoanode exhibited a homogeneous orange transparent film in which no magnetite remained.

In 2014, Leite et al. [11] devised an innovative route for producing nanostructured hematite thin films by coupling the CND method with a magnetic field. This method involved the use of two permanent magnets outside the colloidal magnetite solution to generate a static ferrofluid, and the TCO substrate was deposited via dip coating. Thus, columnar hematite films with a textured orientation in the <110> direction were produced, exhibiting exceptional photoelectrochemical properties [11]. This innovative deposition method was employed to demonstrate that hematite thin films with thicknesses ranging from 30 to 300 nm can be tailored. In contrast to the multiple reported deposition processes for increasing the film thickness, the magnetite colloidal solution technique employs a single deposition and sintering cycle to achieve a thickness of 300 nm [11]. However, the parameters of the CND method employing magnetic nanoparticles aided by a magnetic field to produce hematite nanostructures are not yet fully understood. For example, how the concentration of nanoparticles and the strength of the magnetic field affect the morphological properties of hematite thin films (thickness, degree of texturing, and optical and electrochemical properties) is unclear. Here, we intend to expand on our understanding of this technique and determine the roles of the magnetic field, nanoparticle concentration in the film morphology, and photoelectrochemical performance of the hematite photoanodes.

## 2. Materials and Methods

### 2.1. Synthesis of Magnetite Nanoparticles

Magnetite nanoparticles (Fe_3_O_4_) were synthesized by dissolving 3 mmol of iron acetylacetonate (III) (Aldrich Chemical 99.99%, St. Louis, MO, USA) in 35 mL of oleylamine (Aldrich Chemical 70%) in a three-necked round-bottomed flask (100 mL). The solution was initially heated to 100 °C for 30 min under a vacuum. Subsequently, the temperature was raised to 320 °C for 1 h in a N_2_ atmosphere. After the reaction was complete, the colloidal solution was allowed to cool to room temperature. The magnetite nanoparticles were washed with acetone, ethanol, and isopropyl alcohol. Finally, the nanoparticles were redispersed in toluene, yielding colloidal dispersions with varying concentrations (150, 300, 400, and 500 mg mL^−1^).

### 2.2. Thin Film Preparation

Commercial FTO substrates used in this study were manufactured by Solaronix (F-doped SnO_2_ transparent conductor oxide layer deposited in an aluminum–boron–silicate glass with a typical size of 2 cm × 1 cm) and employed in the film deposition process. Prior to deposition, the FTO substrate was cleaned with soap, isopropyl alcohol, acetone, and toluene and stored in toluene. The colloidal magnetite solution was deposited on FTO by applying dip-coating assisted by a magnetic field, as shown in Figure 1. Two magnets were separated by 2.7 cm, 5.6 cm, and 7.9 cm in a Teflon container, and 1 mL of colloidal solution was loaded into the container (see scheme in Figure 1). The obtained ferrofluid was used to tune the CND process using three different magnetic fields by varying the distance between the two magnets. The magnetic fields were then simulated using COMSOL Multiphysics software [12] to obtain the approximate strength used in the experiments. Round neodymium iron boron magnets with a remanence of 12.1 kG (Magtek, São Caetano do Sul, Brazil, diameter of 25 mm × 10 mm, N35) were used in the simulation. The applied magnetic fields were 5, 15, and 50 mT. The films were produced by controlling the dip-coating parameters, such as immersion speed (60 mm min^−1^) and withdrawal speed (60 mm min^−1^), but not the immersion permanence time. The deposited films were then sintered for 20 min at 850 °C in a tubular furnace. The substrate was then swiftly removed and cooled to room temperature.

### 2.3. Characterization

The thin films were characterized by X-ray diffraction (XRD, Bruker (Billerica, MA, USA), D8 Advance ECO) using CuKα radiation (*λ* = 0.15406 nm), a linear detector LYNXEYE XE (PSD of 2.948°), primary optics (2.5° axial soller and 0.6 mm slit), and secondary optics (2.5° axial soller and 5.4 mm slit). Transmission electron microscopy (TEM) was performed at 300 kV using a JEOL JEM 2100F instrument (Tokyo, Japan). The TEM sample was prepared by in situ milling using a focused ion beam system. The film thickness was determined using field-emission scanning electron microscopy (FESEM; F50 INSPECT, FEI, Hillsboro, OR, USA).

### 2.4. Photoelectrochemical Characterization

Photoelectrochemical measurements were performed in a standard three-electrode cell with the hematite film as the working electrode (0.28 cm^2^ area), Ag/AgCl in a saturated KCl solution as the reference electrode, and a platinum plate as the counter electrode. A 1.0 M NaOH solution (NaOH ACS Aldrich, 99.99%, St. Louis, MO, USA) in highly pure water (pH = 13.6, at 25 °C) was used as the electrolyte. A scanning potentiostat (potentiostat/galvanostat μAutolab III, Metrohm, Herisau, Switzerland) with a 20 mV s^−1^ scan rate was used to measure the dark and illuminated currents. A 250 W ozone-free xenon lamp (Osram, Munich, Germany) and an AM 1.5 filter were used to simulate sunlight (100 mW cm^−2^) (Newport Corp., Irvine, CA, USA).

## 3. Results

### 3.1. Nanoparticles Synthesis

Thermal decomposition of a Fe(acac)_3_ precursor in oleylamine as a solvent and ligand agent produced magnetite nanoparticles. Fe(acac)_3_ decomposition and Fe_3_O_4_ nanoparticle nucleation occurred at 170 °C [13]. The presence of excess oleylamine stimulated the formation of a strong reducing environment, which was sufficient to partially reduce the Fe^3+^ cations to Fe^2+^, resulting in the formation of magnetite. XRD data were easily classified as cubic *fcc* (inverse spinel structure, $Fd\bar{3}m$ (Appendix A), based on magnetite norms (JCDPS Card no. 19-0629) [14]. Rietveld refinement (Appendix A) was performed utilizing the TOPAS software 5.4 (Bruker), yielding a refined lattice parameter of a = 8.378 Å, which is close to the observed standard parameter in the literature (a = 8.396 Å). Comparatively, the maghemite structure (a = 8.3461 Å, JCDPS Card no. 39-1346), which contains only Fe^2+^ cations, has a lower lattice parameter than the magnetite nanoparticles synthesized in this study. This result indicated the presence of both Fe^2+^ and Fe^3+^ in the formation of magnetite. According to the literature, oleylamine acts as a reducing agent for Fe^3+^ cations but is very effective at forming crystalline structures in the presence of iron in two valence states, as determined by XRD [14].

Transmission electron microscopy (TEM) was used to analyze the nanoparticle size, size distribution, structure, and morphology. Figure 2a,b display micrographs and size distribution histograms depicting the formation of nanoparticles with uniform morphology and diameter of approximately 7.6 nm $\left(D_{TEM}\right)$. The selected area electron diffraction (SAED) patterns (Figure 2c) reveal diffraction rings typical of polycrystalline nanoparticles, which correspond to crystallographic planes indexed with Miller indices (*hkl*) of the magnetite phase. This result confirms the XRD measurements of magnetite nanoparticles. For a comparison with $D_{XRD\_r}$, the nanoparticle diameter $D_{TEM}$ must be interpolated using a number-weighted log-normal distribution. The volume-weighted mean diameter $D_{XRD\_v}$ was computed according to Equation (1), as the XRD-obtained particle size was derived from volumetric diffraction data [15].
(1)$$D_{TEM\_V}=D_{TEM}e^{3{\left(\sigma_{D_{TEM}}\right)}^{2}}\;$$
where, $\sigma_{D_{TEM}}$ is the standard deviation of the log-normal distribution. Then, $D_{TEM\_V}$ = 7.9 nm was calculated and compared to $D_{XRD\_r}$ (7.8 nm), revealing an error of less than 2% between the two diameters, correlating the morphological $D_{TEM}$ with the volumetric$\;D_{XRD}$. This comparative result demonstrates that the magnetite nanoparticle region is predominantly made up of single-crystalline particles, i.e., each nanoparticle has a low crystallographic defect concentration and could be referred to as a nanocrystal. 

Figure 2d depicts the qualitative magnetic properties of the colloidal nanoparticles, which indicate ferrofluid formation. In the presence of a magnetic field, the spikes observed in the colloidal solution indicate classical ferrofluid behavior, in which superparamagnetic nanoparticles exhibit a preferred magnetic orientation along the magnetic field lines. Small nanoparticles with diameters smaller than the superparamagnetic radius (<70 nm) lack the remanescence field required for a stable colloidal solution of Fe_3_O_4_ nanoparticles in toluene [16]. In the presence of an externally applied magnetic field, these nanoparticles exhibit a rapid magnetic response by aligning themselves with the field and exhibiting their own magnetism. This behavior was used to tune the preparation of thin films using the dip-coating method. The viscosity of a ferrofluid solution increases in the presence of a magnetic field. This process occurs as a result of the nanoparticles’ partially aligned magnetic moments. In contrast, when there is no magnetic field, the nanoparticles exhibit random Brownian motion [17,18,19].

### 3.2. Nanoparticle Deposition

The morphology and thickness of the hematite thin films were analyzed using scanning electron microscopy (SEM) (Figure 3 and Appendix A). Using the CND method with varying magnetic fields and nanoparticle concentrations, the thickness of thin films could be modified. The cross-sectional and top-view images depict elongated hematite grains with open porosity, also known as mesoporous films, and a range of thicknesses between 60 and 370 nm. By adjusting the magnetic field and concentration of the colloidal solution, the typical appearance of columnar grains was achieved. Figure 3 demonstrates that, according to SEM images, increasing the nanoparticle concentration led to an increase in thickness, while increasing the magnetic field led to a decrease in thickness. The thickness of the thin films is directly proportional to the particle concentration and inversely proportional to the magnetic field strength applied during deposition. During the deposition processes, the thickness and mesoporous formation varied due to the presence of different magnetic fields. At lower concentrations and higher magnetic fields, pores of 10–30 nm were observed, whereas, at higher concentrations and lower magnetic fields, these pores grew to 30–50 nm. Clearly, the magnetic field and concentration have an effect on grain growth and thickness. With an increase in the magnetic field, samples synthesized with 500 mg mL^−1^ exhibited thinner thicknesses and smaller pore sizes. This behavior can be rationally explained by the viscosity of the solution, which decreases in the presence of a weaker magnetic field. However, an excessive increase in viscosity caused by a strong magnetic field tends to keep nanoparticles in the area where the field acts, resulting in a decrease in thickness for stronger magnetic fields. For photoelectrochemical applications, the thickness of the film can be viewed as a crucial factor. Freitas et al. [20] correlated the different thicknesses of hematite obtained by the hydrothermal method, demonstrating that the influence of surface modification promotes polarized states, improves surface trapping, and consequently decreases the lifetime of photogenerated charge. The APCVD method described by Gratzel et al. [21] demonstrates the formation of films with varying thicknesses based on deposition time, as well as the effect of thickness on the low hematite diffusion length (5 nm). The use of a dopant to enhance the film’s electronic transport is a possible solution to this issue.

Due to the significance of film thickness for PEC performance, the correlation between the magnetic field and the growth of thin films was analyzed. The relationship between concentration, magnetic field, and film thickness is depicted in Figure 4. The relationship between film thickness and applied magnetic field (*B*) for a fixed solution concentration is linear: thickness=a−b B. Additionally, fixing the magnetic field at different concentrations results in a nonlinear behavior, wherein the $thickness=a+b{\left[C\right]}^{d}$, where *a*, *b*, and *d* are fitted constants. The linear regression with the negative angular coefficient demonstrated in Figure 4a can be readily observed by the inversely proportional behavior of the thickness with applied *B*. The nonlinear regression presented in Figure 4b indicates that the thickness is proportional to the concentration multiplied by a constant d. Several factors, such as solution viscosity, concentration, superficial tension, film take-off speed, and meniscus curvature, govern the thickness of films produced by dip-coating [22,23,24,25]. The scientific literature contains mathematical equations describing the formation of films and their thicknesses based on these variables [22,26,27,28]. However, these dip-coating equations cannot be used because they do not account for the magnetic field’s influence. Due to the influence of the field on film formation parameters such as viscosity, density, and surface tension of the solution, it was not possible to discover a simple equation that describes the results presented here. As shown in Equation (2), a relationship expression can be specified.
(2)$$Thickness\propto \;\frac{{\left[{\mathrm{Fe}}_{3}O_{4}\;NCs\right]}^{\gamma }}{B}$$
where $\left[{\mathrm{Fe}}_{3}O_{4}\;NCs\right]$ is the concentration of the nanoparticle solution, *B* is the applied magnetic field, and *γ* is a coefficient related to the influence of the field on the proportion of nanoparticles present in the solution.

### 3.3. Thin-Film Characterization

For structural analysis of the hematite films, XRD analysis of the thin films was performed; see Appendix A. All samples were confirmed to be indexable for the rhombohedral hematite phase with space group $R\bar{3}c$. On the FTO substrate, characteristic peaks of the cassiterite phase were also observed, with no evidence of other iron oxide phases. In general, the (104) peak is the most intense diffraction peak for hematite particles with random crystallographic facet orientations. However, the scientific literature indicates that hematite nanostructures with preferred orientation in the [110] direction have superior photoelectrochemical properties (higher photocurrent densities) [29,30,31]. Thus, the XRD results are essential for analyzing and establishing the connection between the hematite photoanode response and the observed PEC results.

The crystallite size was estimated using the Scherrer equation based on the full width at half maximum (FWHM) of the (110) peak; see Appendix A. All samples had a crystallite size of approximately 37–43 nm. The insignificant differences between the samples indicate that the magnetic field has little effect on the crystallite size. As previously mentioned, a decrease in the magnetic field favored the coalescence of the grains, as evidenced by the SEM and TEM images [10,32,33].

Figure 5 is a cross-sectional TEM image of the hematite film synthesized at 5 and 50 mT using 500 mg mL^−1^ at 5 and 50 mT. Observable is the active layer formed by the mesoporous hematite and the FTO substrate layer. The HRTEM analysis revealed a preferential growth of hematite in the [110] direction, corroborating the XRD analysis’ findings. The HRTEM images were acquired at the FTO/α-Fe_2_O_3_ interface (areas A and B in Figure 5a and areas C and D in Figure 5b at magnetic fields of 5 mT and 50 mT, respectively). To index the hematite crystallographic orientation, a fast Fourier transform (FFT) was applied to the HRTEM images of areas A, B, C, and D (see insets for each area in Figure 5a,b). The sample obtained under a stronger magnetic field displayed a growth orientation in the direction of [110], whereas region D displayed a growth orientation in the direction of [104] for a weaker magnetic field. Although statistical analyses of the HRTEM images are impractical for a number of FTO/α-Fe_2_O_3_ interfaces, these results indicate that an increase in the magnetic field aids in the orientation of the hematite crystals, which is consistent with Appendix A. The HRTEM images in Figure 6 depict the atomic lattice fringes of hematite, which coincide precisely with the 3D lattice model along the [104] and [110] directions of the rhombohedral hematite, further demonstrating that the hematite thin films consist of (110) and (104) planes.

The optical properties of the hematite thin films were characterized by UV-vis spectroscopy. Appendix A depicts the absorbance spectra of the films obtained at various magnetic fields and concentrations. All samples exhibited absorption up to 600 nm, corroborating the literature-reported bandgap of 2.1–2.2 eV [34]. Using the UV-vis spectra and the irradiance spectrum under a light source with 1.5 AM–100 mW cm^−2^, it is possible to calculate the absorbed photocurrent density $J_{abs}$ under the condition of a 100% quantum yield (each photon generates one electron-hole pair, i.e., one charge carrier), according to the equation:(3)$$J_{abs}=q\int_{350}^{600}f\left(\lambda \right)A\left(\lambda \right)d\lambda$$
where *q* is the electron charge, *f*(*λ*) is the irradiance spectrum of the light source used for the photoelectrochemical measurements, and *A*(*λ*) is the absorbance spectrum obtained from UV-vis spectroscopy.

The $J_{abs}$ values of hematite thin films are illustrated by comparing the effects of the magnetic field (Figure 7a), the concentration of the colloidal solution (Figure 7b), and the maximum theoretical $J_{abs}$ (Figure 7c) [4]. According to the SEM and TEM analyses, an increase in magnetic field causes a decrease in hematite thickness, and it is anticipated that $J_{abs}$ will decrease as the magnetic field increases. Due to the quantity of material deposited on the FTO substrate, this behavior occurred. Figure 7d depicts the correlation between thickness, $J_{abs}$ and magnetic field, demonstrating that there is an inversely proportional relationship between them, i.e., an increase in the magnetic field causes a decrease in the thickness of the hematite films and their $J_{abs}$ values. This correlation is significant because greater thicknesses and $J_{abs}$ necessitate greater amounts of photoactive material deposition.

The process of forming thin films in the presence of higher magnetic fields promotes a higher-density nanoparticle film (compacted), and the process of sintering and removing the organic ligand can increase the porosity of the film, a factor associated with a decrease in $J_{abs}$.

Appendix A depicts the photocurrent densities (V curves) obtained under front and rear illumination. The relationship between the linear sweep measurement of the thin film at 1.23 V RHE (Reference Hydrogen Electrode) and the magnetic field and concentration under front illumination is depicted in Figure 8a. In comparison, the performance under back illumination was marginally higher than under front illumination, which may be attributable to bulk recombination and electron transport issues [33]. The magnetic field and nanoparticle concentration have a clear effect on the photoelectrochemical outcomes. The use of stronger magnetic fields increases the *J_PH_* values, as depicted in Figure 8a, which reveals that higher concentrations can be observed when using stronger magnetic fields. This result can be attributed to the proportion of the deposited material that is affected by the presence of a magnetic field, which affects both the absorption values and photoelectrochemical results of the film. For pure hematite, a stronger magnetic field promotes the formation of thinner films, resulting in greater photocurrent density values. When the films begin to thicken, photocurrent is diminished. This could be explained by the higher rate of charge recombination and low electron mobility [34,35,36,37]. In contrast, different nanoparticle concentrations also affect the photocurrent density, with *J_PH_* decreasing in the case of higher nanoparticle concentrations in lower magnetic fields. We observed that optimal performance was achieved with a lower concentration deposition or greater magnetic fields. To obtain thicker films with higher nanoparticle concentrations, a stronger magnetic field must be applied. Conversely, at lower concentrations, weaker magnetic fields can be used.

Figure 8b illustrate the correlation between the efficiency of thin films and the magnetic field and concentration under front illumination. Using Equation (4), the global efficiency of the thin-films $\left(\eta_{global}\right)$ was calculated.
(4)$$J_{PH}=J_{abs}\times \eta_{global}\;\;\;\;;\;\;\eta_{global\left(\%\right)}=\frac{J_{PH}}{J_{abs}}\times 100\;$$

The relationship between the efficiency’s dependence on the magnetic field and concentration and the photocurrent density is essentially identical. The magnetic field had a two-step effect on the performance of hematite thin films. At higher magnetic fields, all thin films exhibited a similar efficiency of approximately 7.9%, whereas the film with the lowest concentration achieved an efficiency of 9.3%. In contrast, for lower magnetic fields, the efficiency of these films was highly dependent on their concentration during preparation. The efficiency of the thin films prepared at high concentrations in weaker magnetic fields was poor. To achieve greater efficiency at different nanoparticle concentrations, it is necessary to employ strong magnetic fields; if a weaker magnetic field is employed, lower nanoparticle concentrations are required. Another possible explanation for the effect of magnetism is the presence of a magnetic field that aligns magnetic nanoparticles, which, when the material is converted to hematite, promotes the film’s texturing in the <110> directions, resulting in excellent photoelectrochemical properties. Future research will focus on the dip-coating rise and fall velocity, which could affect the formation of thin films, as well as how the presence of a magnetic field influences this process.

## 4. Conclusions

Our findings indicate that the magnetic-field-assisted CND method produces superior hematite thin-film control and high-performance PEC devices. Moreover, we believe that the magnetic field deposition process is a viable alternative to the active hematite benchmark efficiency. The magnetite nanoparticles that were synthesized were monodispersed with a narrow size distribution, presenting a stable colloidal solution and forming a ferrofluid. A comparison of the film thicknesses in relation to the applied magnetic field demonstrated that the magnetic field intensity could be used to regulate the formation and thickness of these films. In the presence of a stronger magnetic field, the SEM analysis revealed a reduction in the size of the pores. Intriguingly, the PEC results revealed that a stronger magnetic field is required to achieve a higher *J_PH_* and efficiency performance, regardless of the nanoparticle concentration. Applying a weaker magnetic field requires a lower nanoparticle concentration to activate a greater *J_PH_* and efficiency.

## Figures and Tables

**Figure 1 nanomaterials-12-01636-f001:**
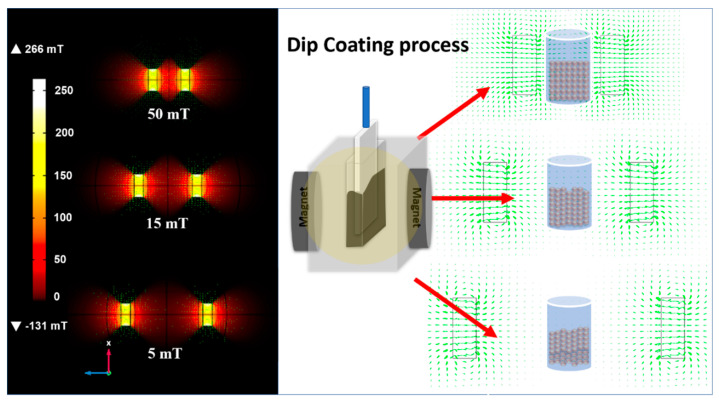
Magnetic field simulation applied in the process of deposition of the magnetite nanoparticles by dip-coating.

**Figure 2 nanomaterials-12-01636-f002:**
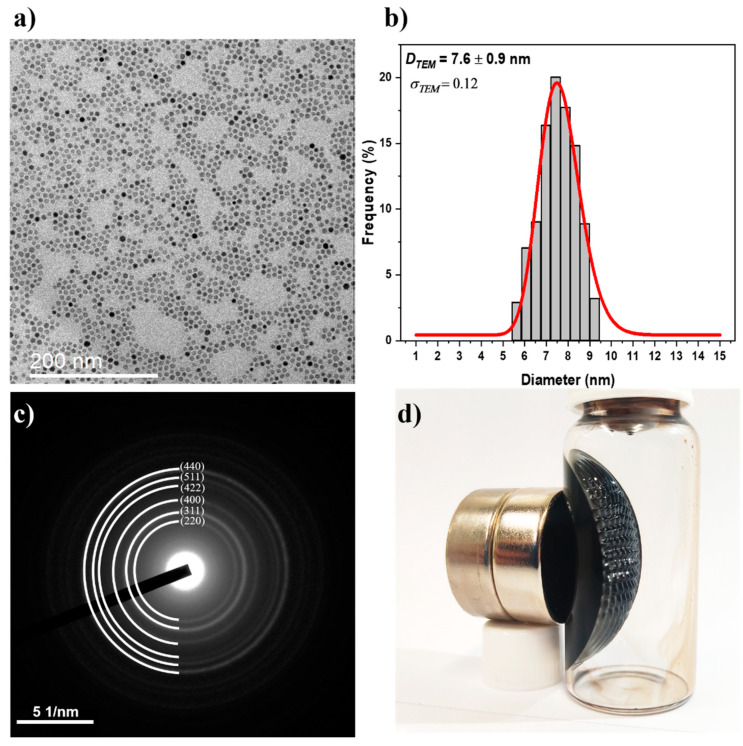
TEM characterization of Fe_3_O_4_ nanoparticles. (**a**) TEM micrograph, (**b**) histogram of size distribution. (**c**) SAED pattern, and (**d**) behavior of nanoparticles in the presence of a magnetic field.

**Figure 3 nanomaterials-12-01636-f003:**
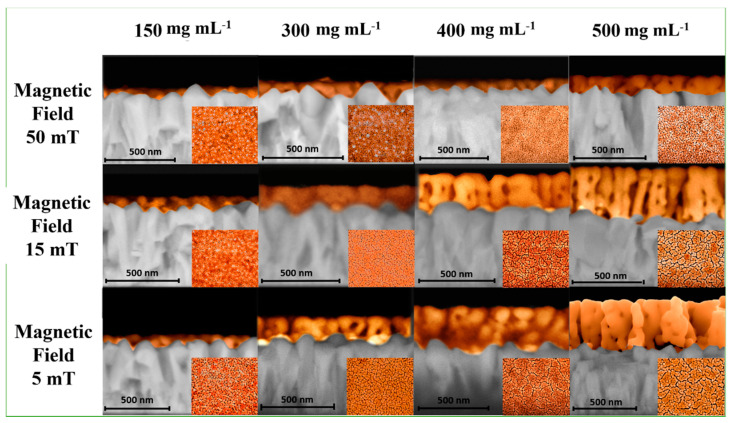
Cross-sectional SEM images and top-view (inset figures) of hematite nanostructures obtained for different magnetic fields and nanoparticle concentrations.

**Figure 4 nanomaterials-12-01636-f004:**
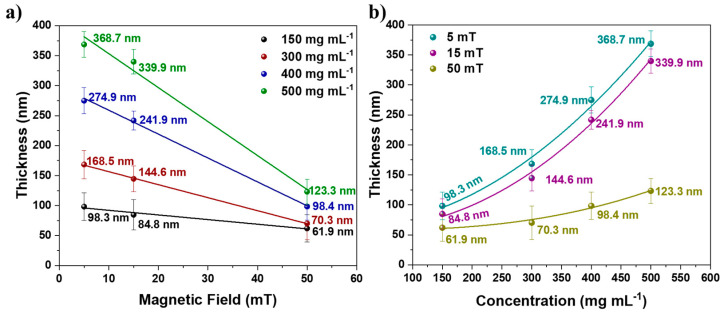
Relationship of film thickness with (**a**) magnetic field and (**b**) nanoparticle concentration.

**Figure 5 nanomaterials-12-01636-f005:**
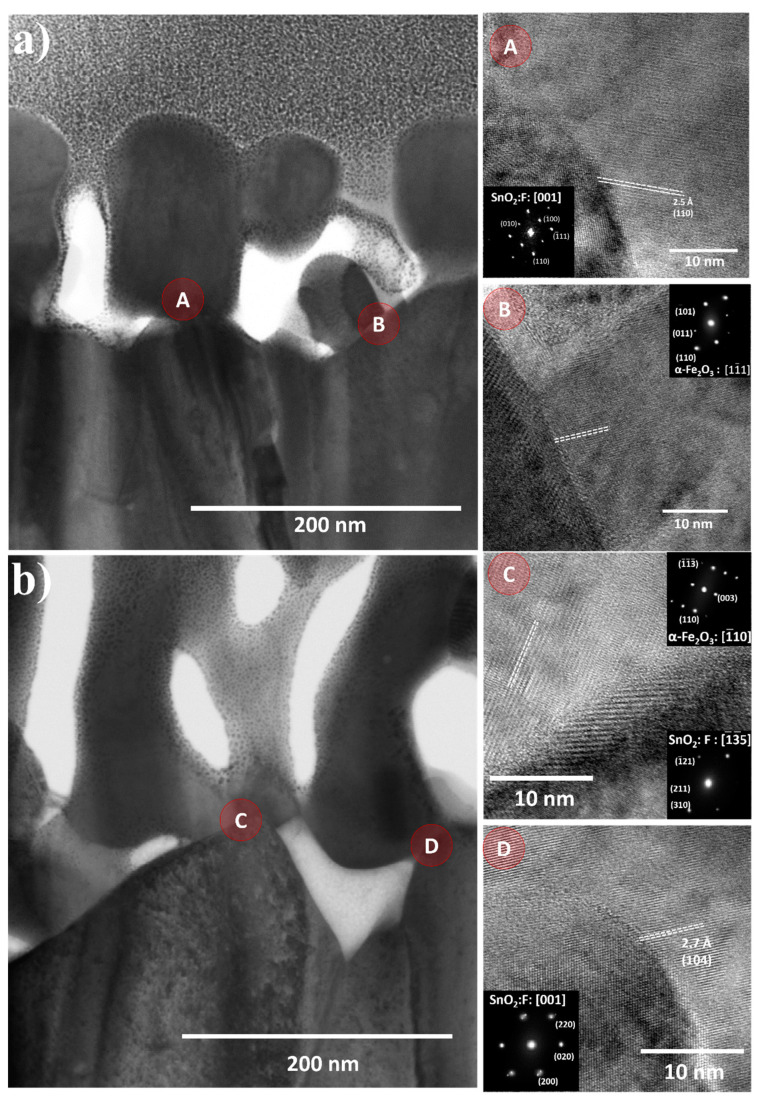
TEM images of the hematite/FTO interface obtained with 500 mg·mL^−1^ at (**a**) 5 mT magnetic field and inset show HRTEM images and Fast Fourier Transform images of hematite and FTO (SnO_2_) indexed with the crystallographic planes along the zone axis of points A and B. (**b**) 50 mT magnetic field, and inset show HRTEM images and Fast Fourier Transform images of hematite and FTO (SnO_2_) indexed with the crystallographic planes along the zone axis of points C and D.

**Figure 6 nanomaterials-12-01636-f006:**
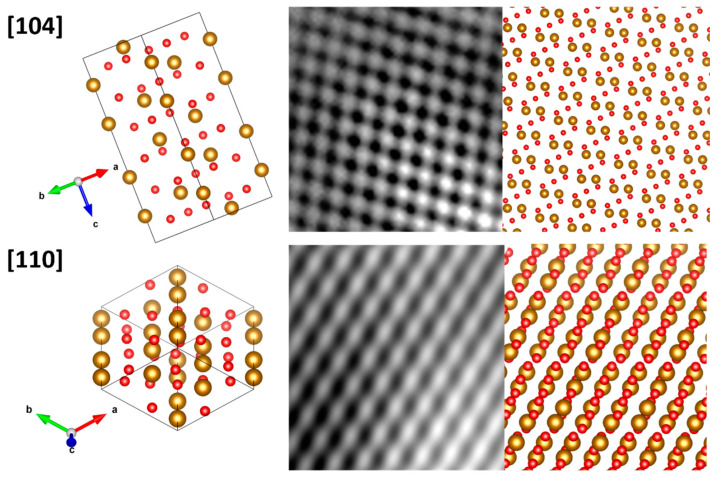
HRTEM images showing the atomic lattice fringes of hematite, which precisely coincide with the 3D lattice model along the [104] and [110] directions of rhombohedral hematite.

**Figure 7 nanomaterials-12-01636-f007:**
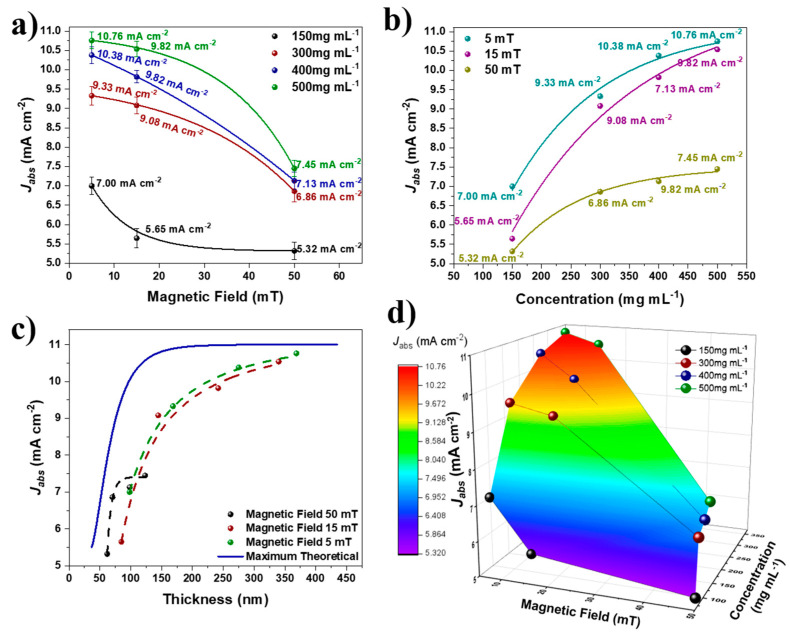
Relationship of film *J_abs_* with: (**a**) magnetic field; (**b**) nanoparticle concentration; (**c**) thickness of hematite films; and (**d**) graph of response surface showing the relationship of magnetic field, thickness, and *J_abs_*.

**Figure 8 nanomaterials-12-01636-f008:**
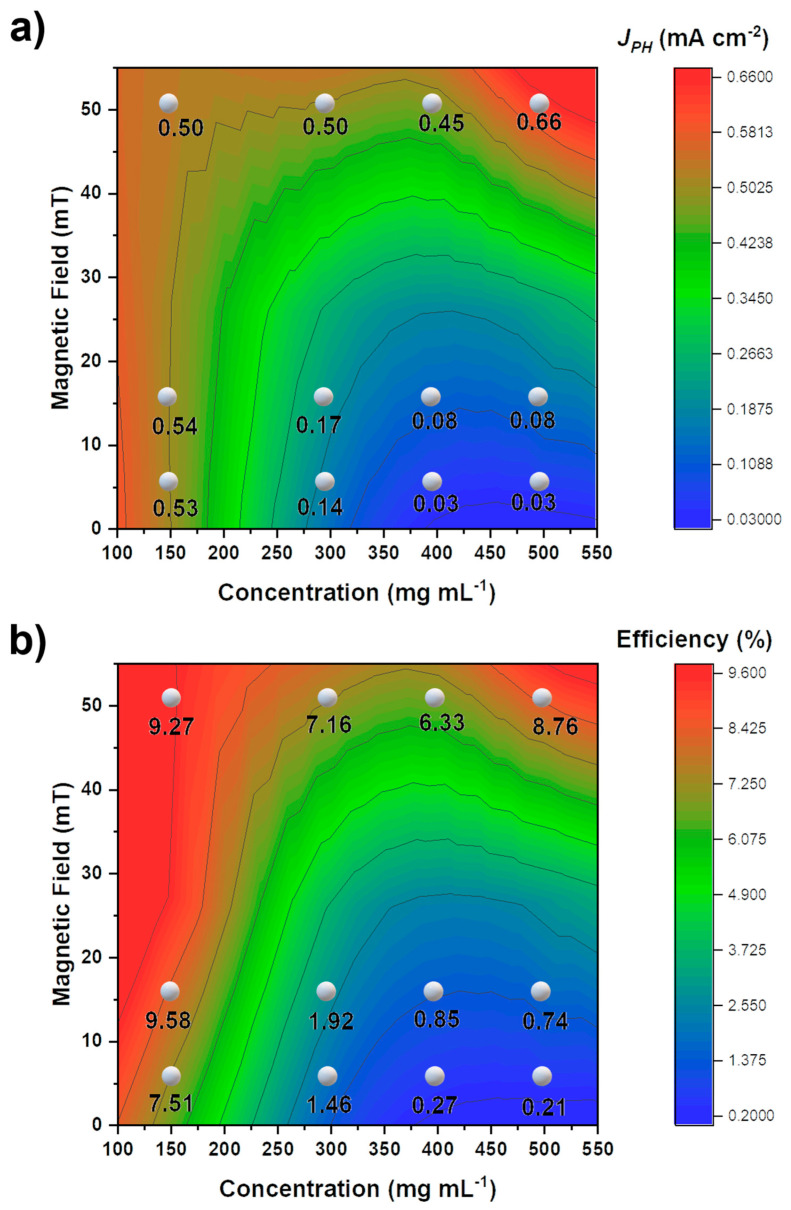
Response of photoelectrochemical measurements for (**a**) front illumination and (**b**) efficiency for different magnetic fields and nanoparticle concentrations.

## Data Availability

The data are included in the article.

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
