# Peer review of "The Influence of Magnetic Field and Nanoparticle Concentration on the Thin Film Colloidal Deposition Process of Magnetic Nanoparticles: The Search for High-Efficiency Hematite Photoanodes"

_nanomaterials, 2022, doi:10.3390/nano12101636_

Round 1

Reviewer 1 Report

This manuscript reported the magnetic field-assisted CND process leads to an outstanding hematite thin films control and high-performance PEC device. This is a very interesting work and can be considered for publication in nanomaterials. The following comments should be addressed;

  1. Some language errors, expression errors, expression errors and so on need to be corrected. For example, in Figure 2a “mg ml-1” should be “mg mL-1”.
  2. The abstract quality should be improved. In the abstract section, the preface expresses too much and does not express the experimental content of the article.
  3. Does the ratio of Fe3+ to Fe2+ in Fe3O4 affect the performance?
  4. The effect of magnetism on properties should be discussed in more detail.
  5. One related literature about PEC should be introduced, Materials Today Sustainability 18 (2022) 100118.

Author Response

  1. Some language errors, expression errors, expression errors and so on need to be corrected. For example, in Figure 2a “mg ml-1” should be “mg mL-1”.

All language errors and expression errors have been corrected as required.

  1. The abstract quality should be improved. In the abstract section, the preface expresses too much and does not express the experimental content of the article.

The sentence has been changed.

  1. Does the ratio of Fe3+ to Fe2+ in Fe3O4 affect the performance?

The material that has a photoactive response and, in turn, a good photoelectrochemical performance is α-Fe2O3 (hematite), and as in this study, obtaining this material is based on the conversion of Fe3O4 (magnetite), the present amount of Fe2+ and Fe3+ influences the obtaining of hematite, and consequently its properties, and the presence of Fe2+ in greater quantity facilitates the conversion process to hematite.

  1. The effect of magnetism on properties should be discussed in more detail

This point has been more discussed as required.

  1. One related literature about PEC should be introduced, Materials Today Sustainability 18 (2022) 100118.

The article recommended has been introduced.

Reviewer 2 Report

The manuscript discusses the effect of magnetic field strength and colloidal magnetite nanoparticle concentration on the efficiency of a hematite Photoanode electrode produced by the Thin Film Colloidal Deposition Process. It is a useful work accompanied by beautiful and informative figures. It is undermined though by a general sloppiness. English language and style require extensive improvement. The authors did not even do a typical spell check. I strongly suggest the MDPI English editing service or a professional translator. Page 2 Lines 45-47 “Also, the crystallographic structure of iron oxide nanoparticles can be tuned from the Fe3O4 (Fe+2O. Fe2+3O3), magnetite) to the γ-Fe2O3 (maghemite) structure.” The transformation of Fe3O4 to Fe2O3 is a chemical reaction and not a “tuning” of the crystallographic structure Page 2 Lines 85-86 “The magnetite nanoparticles were washed with acetone, ethanol and isopropyl. Something is missing. I assume it is isopropyl alcohol. The vendor of the commercial FTO substrate is needed. In the 3.1 section, (Page 4 lines 130-131) the authors claim “Magnetite nanoparticles was synthesized by thermal decomposition of Fe(acac)3 precursor in oleylamine and oleic acid as solvent and ligand agents.” Yet they are not mentioning any addition of oleic acid in the “Synthesis of magnetite nanoparticles” section! SAED and APCVD abbreviations need definition All figures in the supporting information are important and should be incorporated into the main text.

Author Response

All the considerations have been corrected and changed, except the incorporations of the figures present in the supporting information, due all figures presented in the supporting information have been used to construct the figures presented in the main text, made then the incorporation changes the focus of the paper, and finally, the English editing service has been made.

Round 2

Reviewer 2 Report

The manuscript is improved. English language and style still need corrections. 

Author Response

The text has been submitted for English language and style corrections.